# Partition-Based Active Learning for Graph Neural Networks

**Jiaqi Ma**[*]                                                                                          *jiaqima@illinois.edu*
*School of Information Sciences*
*University of Illinois Urbana-Champaign*

**Ziqiao Ma**[*]                                                                                         *marstin@umich.edu*
*Department of Computer Science and Engineering*
*University of Michigan*

**Joyce Chai**                                                                                             *chaijy@umich.edu*
*Department of Computer Science and Engineering*
*University of Michigan*

**Qiaozhu Mei**                                                                                             *qmei@umich.edu*
*School of Information*
*Department of Computer Science and Engineering*
*University of Michigan*

**Reviewed on OpenReview:** *https://openreview.net/forum?id=eOxaRylNuT*

## Abstract

We study the problem of semi-supervised learning with Graph Neural Networks (GNNs) in an active learning setup. We propose GraphPart, a novel partition-based active learning approach for GNNs. GraphPart first splits the graph into disjoint partitions and then selects representative nodes within each partition to query. The proposed method is motivated by a novel analysis of the classification error under realistic smoothness assumptions over the graph and the node features. Extensive experiments on multiple benchmark datasets demonstrate that the proposed method outperforms existing active learning methods for GNNs under a wide range of annotation budget constraints. In addition, the proposed method does not introduce additional hyperparameters, which is crucial for model training, especially in the active learning setting where a labeled validation set may not be available.

## 1 Introduction

Recently, Graph Neural Networks (GNNs) (Kipf & Welling, 2017; Hamilton et al., 2017; Wu et al., 2019a; Veličković et al., 2018) have been drawing increasing research interest, due to its numerous successes in various applications. In particular, GNNs have shown superior performance in *Graph-based Semi-Supervised Learning* (GSSL), a family of problems where individual samples are inter-connected by an underlying graph, and GNNs provide a flexible method to leverage the information stored in the graph to assist learning.

In this paper, we investigate the problem of GSSL with GNNs in an *active learning* setup (Settles, 2012), where one is allowed to actively query node labels on the graph given a limited annotation budget. The active learning setup is particularly interesting in the context of GSSL as we usually have access to abundant unlabeled samples prior to learning and, in many cases (*e.g.*, on a social network), we have the flexibility to query the labels for a small portion of the samples. Furthermore, since a key advantage of GNN is the ability to utilize the relational information among the inter-connected samples, properly selecting nodes to annotate may further enhance GNN's performance.

---

[*]Equal contribution.

However, directly adapting conventional active learning methods to GSSL may be sub-optimal due to the special structure of the problem and the GNN models. Indeed, utilizing proper smoothness properties of the data has been a key ingredient for the success of many active learning methods. For example, a commonly used assumption (which we call *feature smoothness*) is that samples with similar features have higher chances to fall into the same class. In addition to feature smoothness, real-world GSSL tasks often leverage multiple types of smoothness properties over the graph, spanning the spectrum between *local smoothness* and *global smoothness* (Zhou et al., 2004). Local smoothness usually relies on the pairwise Euclidean distance between node representations, which becomes less informative in high-dimensional space (Aggarwal et al., 2001). Global smoothness, on the other hand, is a commonly seen phenomenon in graph-structured data and serves as a great supplement to the local smoothness. While there exist graph-based active learning methods utilizing some of these smoothness properties (Gu & Han, 2012; Ji & Han, 2012; Dasarathy et al., 2015; Cai et al., 2017; Wu et al., 2019b), methods that fully utilize feature and structural smoothness at the proper level are rare.

We propose **GraphPart**, a Graph-Partition-based active learning method for GNNs. The method is largely motivated by the community structures that are commonly present in real-world graphs. Node and structural properties often exhibit homogeneity within a community and heterogeneity across communities. We formalize this observation with proper smoothness assumptions of the graph-structured data at community level (represented by partitions of the graph) and conduct a novel analysis of the GNN classification error under such assumptions. The analysis further motivates the graph-partitioning step in the proposed method, GraphPart. In particular, GraphPart first splits the graph into several partitions based on modularity (Clauset et al., 2004) and then selects the most representative nodes within each partition to query. An important merit of the proposed method is that it does not introduce additional hyperparameters, which is crucial for active learning setups as labeled validation data are often absent. Through extensive experiments, we demonstrate that the proposed method outperforms existing active learning methods for GNNs on multiple benchmark datasets for a wide range of annotation budgets. In addition, the proposed active learning method is able to mitigate the accuracy disparity phenomenon of GNNs (Ma et al., 2021).

## 2 Related Work

Active learning is a sub-field of machine learning that primarily concerns the bottleneck in expensive annotation and attempts to achieve high accuracy by querying as few labeled samples as possible. Early efforts in this field prior to the emergence of deep learning have been comprehensively summarized by (Settles, 2012).

**Active Learning Setups.** The classic active learning algorithms query one sample at a time and label it. Such a setting is inefficient for a deep learning model as it frequently retrains but updates little, and it is prone to overfitting (Ren et al., 2020). Therefore, in deep active learning, the batch-mode setting, where a diverse set of instances are sampled and queried, is more often considered. In recent years, the *optimal experimental design* principle (Pukelsheim, 2006; Allen-Zhu et al., 2020) motivates the machine learning community to minimize the use of training resources and avoid tuning on a validation set. Combining the settings of one-shot learning and batch-mode active learning, some recent studies (Contardo et al., 2017; Wu et al., 2019b) adopt a one-step batch-mode active learning setting. In each run, the algorithm uses up the predefined budget to select a batch of nodes to label. The querying process is done once and for all in order to minimize retraining. We focus on such one-step batch-mode setup as we concern with active learning for graph neural networks.

**Active Learning on Graphs.** Early works in active learning on graphs (Bilgic et al., 2010; Gu & Han, 2012; Ji & Han, 2012; Dasarathy et al., 2015) are designed specifically for non-deep-learning models and/or fail to take the node features into consideration. As deep geometric learning and GNNs become popular, iterative node selection criteria were designed upon the expressiveness of GNNs. In particular, AGE (Cai et al., 2017) evaluates the informativeness of nodes by linearly combining centrality, density, and uncertainty, (Gao et al., 2018) further proposed ANRMAB, which extends this framework by dynamically learning the weights with a multi-armed bandit mechanism and maximizing the surrogate reward. Similarly, (Chen et al., 2019) follows previous works and proposed ActiveHNE, which extends AL on non-i.i.d data and heterogeneous networks. The above frameworks neglect interaction between nodes and suffer from sub-optimality due to short-term surrogate criteria. Upon this observation, (Hu et al., 2020a) proposes GPA, which formalizes the AL task

as a Markov decision process and learns the optimal query strategy with reinforcement learning techniques. More recent works approached the problem by incremental clustering (Liu et al., 2020), and adversarial learning (Li et al., 2020). Yet, all of these models are based on an iterative setting. Given some efforts to avoid validation (Regol et al., 2020), iterative querying and retraining is still required.

Another line of optimization-based approaches develops active learning algorithms by investigating the upper bounds of the classification loss. FeatProp (Wu et al., 2019b) is one of the state-of-the-art active learning approaches for GNNs, which derives an upper bound of the node classification error under smoothness assumptions over the label distributions and GNN models. The theoretical analysis by (Wu et al., 2019b) follows prior work, including ANDA (Berlind & Urner, 2015) which proves a finite sample bound on the expected loss of KNNs in the covariate shift setting, and Coreset (Sener & Savarese, 2018) which conducts an analysis on Convolution Neural Networks. A common pattern of the active learning algorithms along this line is that the algorithms seek to select a set of nodes that achieves a good coverage of the space of sample features or hidden representations, and this is typically done via certain clustering algorithms. Our work falls into this category. Compared to existing methods, our method utilizes both the local and global smoothness properties of graph-structured data, where the latter was largely missing in the literature on active learning for GNNs so far.

## 3 Preliminaries

### 3.1 Notations

We start by introducing useful notations to characterize node classification with GNNs.

**Attributed Graph** The task of node classification is defined on an attributed graph. Define $[n] = \{1, 2, \cdots, n\}$. We denote a graph of size $n$ as $G = (V, A)$, where $V = [n]$ is the set of nodes and $A \in \{0, 1\}^{n \times n}$ is the adjacency matrix. Each node $i \in V$ is associated with a $d$-dimensional feature vector $\mathbf{x}_i \in \mathcal{X} \subseteq \mathbb{R}^d$, and a label $y_i \in \mathcal{Y} = [C]$, where $C$ is the number of classes. Denote $X \in \mathbb{R}^{n \times d}$ as the feature matrix stacking each node feature. Given the original adjacency matrix $A$, the degree matrix $D \in \mathbb{R}^{n \times n}$ is defined as $D_{ii} = \sum_j A_{ij}, \forall i \in V$, and $D_{ij} = 0, \forall i, j \in V, i \neq j$. We denote $S$ as the normalized adjacency matrix with added self-loops: $S = (I + D)^{-\frac{1}{2}}(A + I)(I + D)^{-\frac{1}{2}}$.

**Graph Neural Networks** GNNs are a family of neural networks modeling graph-structured data. A GNN model is typically defined with two types of operations on the node representations: aggregation and transformation. For example, an $L$-layer Graph Convolution Network (GCN) (Kipf & Welling, 2017) can be recursively represented by a series of aggregation operations $g_{\text{GCN}}^{(l)}$ and transformation operations $h_{\text{GCN}}^{(l)}$, for $l = 1, \ldots, L$, defined as follows,

$$\widetilde{H}^{(l)} = g_{\text{GCN}}^{(l)}(H^{(l-1)}, G) := SH^{(l-1)}, \quad \text{and}$$

$$H^{(l)} = h_{\text{GCN}}^{(l)}(\widetilde{H}^{(l)}) := \text{ReLU}(\widetilde{H}^{(l)}W^{(l)})$$

where $H^{(0)} = X$ is the feature matrix, $\text{ReLU}(\cdot)$ is the element-wise rectified-linear unit activation function (Nair & Hinton, 2010), and $W^{(l)}$ is the parameter matrix for the transformation operation $h_{\text{GCN}}^{(l)}$.

Noticing that, in most GNNs, the prediction for each node only depends on a local neighborhood of that node, we can define a general GNN as a function that maps a local neighborhood of a node to its classification predictions. Define $\mathcal{N}^{(L)}(i)$ as the set of all neighboring nodes of node $i$ that can be reached within $L$-hop on the graph (including itself), and $G_i^{(L)}$ as the subgraph of $G$ restricted on $\mathcal{N}^{(L)}(i)$. For each node $i$, define $v_i^{(L)} := \left(\{x_j\}_{j \in \mathcal{N}^{(L)}(i)}, G_i^{(L)}\right)$, which is a tuple including all the node features and the subgraph in node $i$'s local neighborhood. Let $\mathcal{V}^{(L)}$ be the space of such possible $v_i^{(L)}$. Then we can define a GNN as a function

$$f : \mathcal{V}^{(L)} \to \mathbb{R}^C.$$

In our later analysis in Section 4.1, we will consider a type of abstract GNNs in the form of $f = h \circ g$, where the aggregation operation $g : \mathcal{V}^{(L)} \to \mathbb{R}^{d'}$ and the transformation operation $h : \mathbb{R}^{d'} \to \mathbb{R}^C$ are separated. In

particular, $g$ can represent any types of feature aggregation on the $L$-hop ego-network of a node. And $h$ is a transformation function with learnable parameters, which is often modeled by a Multi-Layer Perceptron (MLP). Some recently developed GNN models, such as SGC (Wu et al., 2019a) and APPNP (Klicpera et al., 2019), are special cases of this form.

In the rest of our paper, we omit the superscription $(L)$ in notations for simplicity.

**Margin Loss**   Given some $\gamma \geq 0$, the *margin loss* (Neyshabur et al., 2018) on a GNN classifier $f : \mathcal{V} \to \mathbb{R}^C$ for a given labeled sample $(v_i, y_i)$ is defined as follows,

$$\mathcal{L}_\gamma(f(v_i), y_i) := \mathbb{1}\big[f(v_i)[y_i] \leq \gamma + \max_{c \neq y_i} f(v_i)[c]\big], \tag{1}$$

where $\mathbb{1}(\cdot)$ is the indicator function. When we set $\gamma = 0$, $\mathcal{L}_0$ corresponds to the 0-1 classification error.

### 3.2   Active Learning for Graph Neural Networks

In this work, we focus on the one-step batch-mode active learning setup (Contardo et al., 2017; Wu et al., 2019b) for node classification using GNNs. Suppose there is an attributed graph where the graph structure and the node features are known but the node labels are unobserved. We assume that for each node $i$, the node label $y_i$ is a random variable following a conditional distribution $\Pr[y_i \mid v_i]$.

At the beginning of learning, we are given a small set $\mathbf{s}_0 \subseteq V$ of nodes being labeled, which we call a *seed set*. Given all the node samples $\{v_i\}_{i \in V}$ and labels on the seed set $\{y_i\}_{i \in \mathbf{s}_0}$, for a fixed annotation budget $b > 0$, an active learning algorithm aims to carefully select a set of nodes $\mathbf{s}_1 \subseteq V \setminus \mathbf{s}_0$ and $|\mathbf{s}_1| \leq b$, query the labels on $\mathbf{s}_1$, and train a GNN model $\hat{f}$ based on the labeled data $\{(v_i, y_i)\}_{i \in \mathbf{s}_0 \cup \mathbf{s}_1}$, such that the expected classification error on the remaining unlabeled set is small.

## 4   A Graph-Partition-Based Active Learning Framework

The key motivation of the proposed method is that many real-world graphs present community structures, which induces a proper level of smoothness properties in the graph-structured data. In this section, we first formalize this motivation through an analysis of the GNN performance given proper smoothness assumptions, and then introduce the proposed graph-partition-based active learning approach.

### 4.1   An Analysis of Expected Classification Error Under Smoothness Assumptions

The main result of our analysis is an upper bound (Proposition 1) on the quantity $\mathbb{E}_{y_i} \mathcal{L}_0(f(v_i), y_i)$, the expected classification error for a node $i$ and a fixed GNN model $f$.

**Assumptions**   We first state the set of assumptions in order to establish the upper bound. Denote a $K$-*partition* of the graph as $\mathcal{T}_K = \{T_1, T_2, \cdots, T_K\}$, where $T_1, \ldots, T_K \subseteq V$ are disjoint partitions satisfying $\bigcup_{k=1}^K T_k = V$. Recall that we consider GNNs in the form of $f = h \circ g$, where $g : \mathcal{V} \to \mathbb{R}^{d'}$ is a feature aggregation function and $h : \mathbb{R}^{d'} \to \mathbb{R}^C$ is an MLP that takes the aggregated features as input and output the classification logits. We make the following two assumptions on the smoothness of the label distribution and the model.

**Assumption 1** (Label Smoothness). *Assume that $\forall c \in [C]$, there exists a function $\eta_c : \mathcal{V} \to [0, 1]$ such that $\Pr[y_i = c \mid v_i] = \eta_c(v_i)$ for any $i \in V$. Moreover, $\forall k \in [K]$, $\forall i, j \in T_k$, assume that there exists a constant $\delta_\eta < \infty$, such that*

$$|\eta_c(v_i) - \eta_c(v_j)| \leq \delta_\eta \|g(v_i) - g(v_j)\|_2.$$

**Assumption 2** (Model Smoothness). *Assume that $\forall e, e' \in \mathbb{R}^{d'}$, the MLP $h$ satisfies $\|h(e) - h(e')\|_\infty \leq \delta_h \|e - e'\|_2$ for some constant $\delta_h < \infty$.*

**The Main Result**  We use $T(i)$ to denote the partition where the node $i$ belongs to, and denote for convenience the training set $S_{tr} := \mathbf{s}_0 \cup \mathbf{s}_1$ and the test set $S_{te} := V \setminus S_{tr}$. We have the following result.

**Proposition 1.** *For any fixed GNN model $f$, under Assumptions 1 and 2, for any $i \in S_{te}$, if $S_{tr} \cap T(i) \neq \emptyset$, letting $\tau(i) := \arg\min_{l \in S_{tr} \cap T(i)} \|g(v_i) - g(v_l)\|_2$, $\varepsilon_i := \|g(v_i) - g(v_{\tau(i)})\|_2$, and $\gamma_i := 2\delta_h \varepsilon_i$, then we have*

$$\mathbb{E}_{y_i}[\mathcal{L}_0(f(v_i), y_i)] \leq C\delta_\eta \varepsilon_i + \mathbb{E}_{y_{\tau(i)}}[\mathcal{L}_{\gamma_i}(f(v_{\tau(i)}), y_{\tau(i)})]. \tag{2}$$

Proposition 1 provides an upper bound of the expected classification loss $\mathbb{E}_{y_i}[\mathcal{L}_0(f(v_i), y_i)]$ for each node $i$ in the test set $S_{te}$. This upper bound is primarily dependent on the training node $\tau(i)$ and their distance $\varepsilon_i$ on the aggregated feature space, where $\tau(i)$ is the closest training node to $i$ among the ones residing in the same graph partition as $i$. Specifically, the first term is linearly proportional to $\varepsilon_i$; and the second term (the expected margin loss of $\tau(i)$) is increasing with respect to $\gamma_i$ and hence is also increasing with respect to $\varepsilon_i$. This upper bound motivates an active learning algorithm that selects the training set by minimizing $\sum_{i \in S_{te}} \varepsilon_i$, which we will introduce in detail in Section 4.2.

**Remarks**  We make a few remarks on the bound (2) in comparison to relevant previous work. Our novel technical contributions in the analysis include that 1) we have a weaker assumption on label smoothness; and 2) our proof (Appendix A.1) removes an unrealistic implicit assumption in previous work.

First, our analysis can be viewed as an extension of the results by (Sener & Savarese, 2018) and (Wu et al., 2019b). One key difference between our analysis and theirs lies in the assumption on label smoothness (Assumption 1). Adapting the label smoothness assumption by (Sener & Savarese, 2018) and (Wu et al., 2019b) into our notations gives the following Assumption 3.

**Assumption 3** (Label Smoothness by (Sener & Savarese, 2018) and (Wu et al., 2019b))**.** *Assume that $\forall c \in [C]$, there exists a function $\eta_c : \mathcal{V} \to [0, 1]$ such that $\Pr[y_i = c \mid v_i] = \eta_c(v_i)$ for any $i \in V$. Moreover, $\forall i, j \in V$, assume that there exists a constant $\delta'_\eta < \infty$, such that*

$$|\eta_c(v_i) - \eta_c(v_j)| \leq \delta'_\eta \|g(v_i) - g(v_j)\|_2.$$

The following fact indicates that Assumption 1 is weaker than Assumption 3.

**Lemma 1.** *If Assumption 3 holds, then there exists a constant $\delta_\eta \leq \delta'_\eta$ satisfying Assumption 1.*

Intuitively, our Assumption 1 is weaker than Assumption 3 because the latter requires the smoothness condition to hold on the entire graph, while our Assumption 1 only requires the smoothness condition to hold within partitions of the graph. This difference also has a practical significance as Assumption 1 allows the label distributions to have sharp changes across different partitions of the graph, which is more realistic than Assumption 3 that does not allow such sharp changes. In addition, the $\delta_\eta$ in Assumption 1 could be much smaller than the $\delta'_\eta$ Assumption 3 if the label distribution presents *global smoothness* over the graph (Zhou et al., 2004), i.e., the distributions of node labels tend to be closer for nodes within a partition/community of the graph than for nodes reside in different partitions/communities.

Second, while the upper bounds on classification error given by (Sener & Savarese, 2018) and (Wu et al., 2019b) seem to be tighter than our bound (2), we remark that they made an implicit assumption in the proofs that is counter-intuitive. In particular, the implicit assumption is that, the label distributions of the selected training samples are always concentrated on one class. This assumption is counter-intuitive as the selection of training samples alters the label distributions of those samples. We avoid such an assumption in our analysis at the expense of introducing an extra margin loss term.

Finally, we remark that while GCN cannot be written in the form of $f = h \circ g$ as we assumed in our analysis, it has been shown by (Wu et al., 2019b) that the difference between outputs of an $L$-layer GCN on two nodes $i, j \in V$ can be upper bounded by $\delta_{GCN} \|(S^L X)_i - (S^L X)_j\|_2$ for some constant $0 < \delta_{GCN} < \infty$. So the analysis in this section can be similarly applied to GCN.

## 4.2 The Proposed Graph Partition-Based Active Learning Framework

**The General Framework**  Motivated by Proposition 1, we propose an active learning framework that selects the set of nodes to be labeled, $\mathbf{s}_1$, by solving the following optimization problem: for $\varepsilon_i$ as defined in

---

**Algorithm 1** Graph-Partition-Based Query

---

**Input**: A $K$-partition $\mathcal{T}_K$ of the graph, budget $b$
**Output**: A subset of unlabelled nodes $\mathbf{s}_1$ of size $b$: $\mathbf{s}_1 \subseteq V \setminus \mathbf{s}_0$ and $|\mathbf{s}_1| = b$

1: Set $\mathbf{s}_1 = \emptyset$.
2: **for** $T_k \in \mathcal{T}_K$ **do**
3:      $b_k \leftarrow b // K$.
4:      $T_k \leftarrow T_k \setminus \{\mathbf{s}_0 \cup \mathbf{s}_1\}$.
5:      $E_k \leftarrow \{g(v_i)\}_{i \in T_k}$.
6:      $\mathbf{s} \leftarrow b_k\text{-Medoids}(E_k)$.
     //Perform K-Medoids clustering on the set of data points $E_k$ with $b_k$ medoids returned as s.
7:      $\mathbf{s}_1 = \mathbf{s}_1 \cup \mathbf{s}$.
8: **end for**
9: **return** $\mathbf{s}_1$

---

Proposition 1,

$$\min_{\mathbf{s}_1 : |\mathbf{s}_1| \leq b} \sum_{i \in S_{te}} \varepsilon_i = \min_{\mathbf{s}_1 : |\mathbf{s}_1| \leq b} \sum_{i \in V} \min_{j \in T(i) \cap (\mathbf{s}_0 \cup \mathbf{s}_1)} ||g(v_i) - g(v_j)||_2. \tag{3}$$

For any partition, $T_k \in \mathcal{T}_K$, if we further specify a budget $b_k$ for the number of nodes to be selected from this partition, such that $\sum_{k=1}^{K} = b_k = b$, then we can approximately[1] re-write the optimization problem (3) as $K$ separate optimization problems as follows: for $k = 1, \ldots, K$,

$$\min_{\mathbf{s}_1^{(k)} \in T_k : |\mathbf{s}_1^{(k)}| \leq b_k,} \sum_{i \in T_k} \min_{j \in \mathbf{s}_1^{(k)}} ||g(v_i) - g(v_j)||_2, \tag{4}$$

where the individual optimization problem (4) is equivalent to minimizing the objective of a K-Medoids problem with $b_k$ medoids on the partition $T_k$.

Given a $K$-partition of the graph and a feature aggregation function $g : \mathcal{V} \to \mathbb{R}^{d'}$, we summarize the proposed general framework, which we call it **GraphPart** (Graph-Partition-based query), in Algorithm 1. For the K-Medoids algorithm, in practice, we apply an efficient approximation by (Park & Jun, 2009) by selecting nodes closest to K-Means centers. The aggregation function $g$ should be chosen according to the GNN architecture that we will be training. For example, for a 2-Layer GCN model, we set $g(v_i) = (S^2 X)_i$ for any $i \in V$, since this type of aggregation function effectively reflects the output difference of GCN as we remarked at the end of Section 4.1.

**The Graph-Partition Method** We use a modularity-based graph partition method, which is one of the most popular classes of methods for community detection (Newman, 2004; Clauset et al., 2004). Specifically, we obtain a $K$-partition of a graph using the Clauset-Newman-Moore greedy modularity maximization (Clauset et al., 2004) method. This method is a bottom-up algorithm, which begins with communities containing every single node, and iteratively merges the pair of communities that increases modularity the most, until $K$ communities are left. If no pairs of communities can be merged to increase modularity when more than $K$ communities are left, we further use agglomerative hierarchical clustering (Kaufman & Rousseeuw, 2009) to iteratively merge small outlying communities until only $K$ communities are left. Notably, we also propose a simple elbow method to automatically determine the number of partitions $K$ without the need for node labels. Therefore the whole active node selection process is hyperparameter-free.

**Compensating for the Interference across Partitions** One potential shortcoming of the proposed GraphPart method shown in Algorithm 1 is that it ignores the interference across partitions as it optimizes separate $K$-Medoids problems independently on each partition. The medoids selected on two different partitions may be close to each other, which compromises the overall coverage of the unlabeled nodes. To compensate for this problem, we come up with a greedy correction that when selecting the medoids in

---

[1]Omitting the seed set $\mathbf{s_0}$, which is 0 in the one-step setting.

the partition $T_k$, we penalize the nodes that are close to the medoids already selected in the partitions $T_1, T_2, \ldots, T_{k-1}$. Instead of selecting nodes closest to K-Means centers, the distance function to minimize is penalized by the minimum distance to any selected node. We name this corrected variant of GraphPart as **GraphPartFar**, which makes sure that all the nodes returned to $\mathbf{s}_1$ are not too close and similar to each other, increasing the diversity of the pool.

### 4.3 Complexity Analysis

We provide a complexity analysis as follows. In natural sparse networks, the number of edges $m$ is often linear in the number of nodes $n$. Under this approximation, the complexity of Clauset-Newman-Moore community detection is $O(n \log^2 n)$. The agglomerative hierarchical clustering after Clauset-Newman-Moore community detection can be considered as $O(1)$. Under a budget of $b$ and $d$ hidden channels, the K-Means++ has a complexity of $O(bnd/K^2)$ on each of the $K$ communities, adding up to $O(bnd/K)$. Overall, the complexity of the proposed methods is roughly linear in terms of the number of nodes $n$. Nevertheless, we note that in a one-step active learning setup, the computational complexity of the active learning algorithms is usually not the primary challenge. Instead, the cost of expert annotation is often the primary bottleneck problem.

## 5 Experiments

### 5.1 Experiment Setup

We experiment on 7 benchmark datasets and compare the proposed methods against several state-of-the-art baseline active learning approaches on training 3 different GNN models. Following (Wu et al., 2019b), we evaluate each baseline with a series of label budgets and report the Macro-F1 performance for node classification over the full graph. Each experiment setting is repeated with 10 random seeds.

**Dataset.** We experiment on citation networks Citeseer, Cora, and Pubmed (Sen et al., 2008), three standard node classification benchmarks. We also experiment on Corafull (Bojchevski & Günnemann, 2018) and Ogbn-Arxiv (Hu et al., 2020b), for performance on denser networks with more classes, and on co-authorship networks (Shchur et al., 2018) for diversity. The summary statistics of the datasets are provided in Table 1. The homophily ratio is defined following Zhu et al. (2020). To determine the best number of $K$ communities in the graph, we find the elbow of the costs using the `kneebow` [2] toolkit. The value of $K$ for each dataset is illustrated in Figure 1.

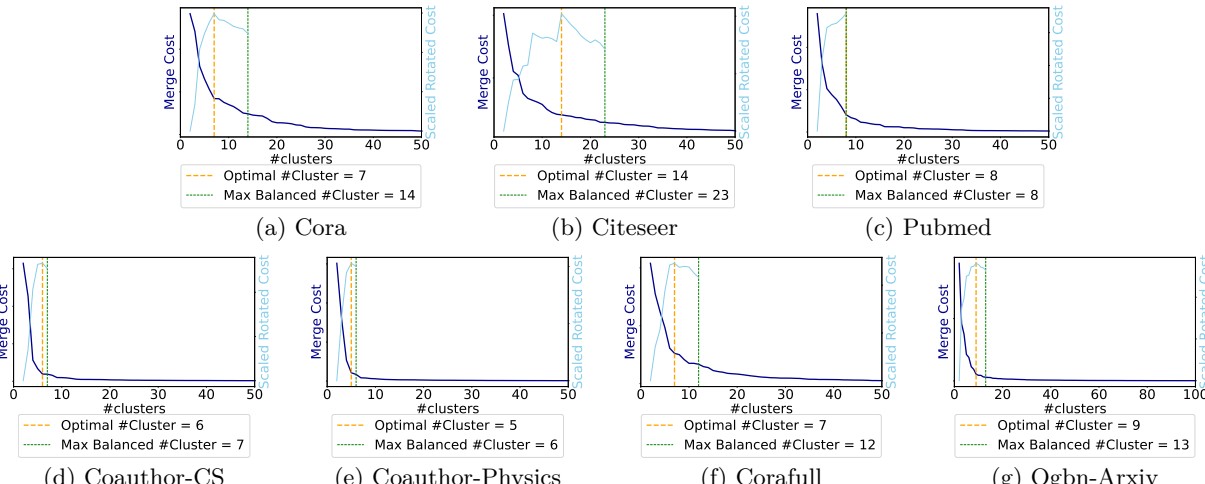

Figure 1: A summary of the number of partitions, automatically determined by the elbow method.

---

[2]https://pypi.org/project/kneebow/

Table 1: Summary statistics of datasets.

| Dataset | #Nodes | #Edges | #Features | #Classes ($C$) | #Partitions ($K$) | Homophily |
|---------|--------|--------|-----------|-----------------|--------------------|-----------|
| Cora | 2,708 | 5,278 | 1,433 | 7 | 7 | 0.810 |
| Citeseer | 3,327 | 4,552 | 3,703 | 6 | 14 | 0.736 |
| Pubmed | 19,717 | 44,324 | 500 | 3 | 8 | 0.802 |
| Co-CS | 18,333 | 81,894 | 6,805 | 15 | 6 | 0.808 |
| Co-Physics | 34,493 | 247,962 | 2,000 | 5 | 5 | 0.931 |
| Corafull | 19,793 | 126,842 | 8,710 | 70 | 7 | 0.567 |
| Ogbn-Arxiv | 169,343 | 1,166,243 | 128 | 40 | 9 | 0.655 |

**GNN Models.** We perform experiments over different popular GNN models, including a 2-layer GCN (Kipf & Welling, 2017) with 16 hidden neurons, a 2-layer GraphSAGE (Hamilton et al., 2017) with 16 hidden neurons and a 2-layer GAT (Veličković et al., 2018) with 8 attention heads with 8 hidden neurons each. For Ogbn-Arxiv dataset, the number of hidden neurons is increased to 128. To train each model, we use an Adam optimizer with an initial learning rate of $1 \times 10^{-2}$ and weight decay of $5 \times 10^{-4}$. As in the active learning setup, there should not enough labeled samples to be used as a validation set, we train the GNN model with fixed 300 epochs in all the experiments and evaluate over the full graph.

**Baselines.** We compare active learning methods that can be applied to the one-step setting, divided into two categories: 1) general-purpose methods that are agnostic to the graph structure (Random, Density, Uncertainty, and CoreSet); and 2) methods tailored for graph-structured data (Centrality, AGE, FeatProp, GraphPart[Far]).

- **Random**: Chooses nodes uniformly at random.

- **Density** (Cai et al., 2017): First performs clustering on the hidden representations of the nodes, and then chooses nodes with maximum density score, which is (approximately) inversely proportional to the $L_2$-distance between each node and its cluster center.

- **Uncertainty** (Settles & Craven, 2008): Chooses the nodes with maximum entropy on the predicted class distribution.

- **CoreSet** (Sener & Savarese, 2018): Pperforms K-Center clustering over the hidden representations of nodes. Since the MIP optimized version is not scalable to large datasets, we use the time-efficient greedy approximation described in the original work.

- **Centrality**: Chooses nodes with the largest graph centrality metric value, which only considers only the graph structure, but not the node features. As is empirically validated by (Cai et al., 2017), **Degree** centrality and **PageRank** centrality outperform other metrics.

- **AGE** (Cai et al., 2017): defines the informativeness of nodes by linearly combining three metrics: centrality, density, and uncertainty. It further chooses nodes with the highest scores.

- **FeatProp** (Wu et al., 2019b): first performs K-Means on the aggregated node features, and then chooses the nodes closest to the cluster centers.

- **GraphPart** and **GraphPartFar**: Two variants of the proposed method as described in Section 4.2.

**Active Learning Setup.** In our one-step setting, we vanish the seed set $s_0$ to zero. Note that some baseline methods (Density, Uncertainty, CoreSet, and AGE) are intended for iterative settings and require an initial model trained with the seed set $\mathbf{s}_0$, as these methods rely on the hidden representations of nodes or the predicted class distribution returned by the initial model. For these baselines, we choose one-third of the budget as random initialization, and let the method select the other two-thirds. On smaller datasets with sparser networks and fewer classes, we test each active learning approach with the series of budgets chosen as $2^{\{0,1,2,3,4\}} \times 10$. On the large datasets where the network is denser and the number of classes is drastically larger, the budgets are chosen as $2^{\{3,4,5,6,7\}} \times 10$.

**Implementation Details.** We adopt PyTorch Geometric (Fey & Lenssen, 2019) for dataset preprocessing and model construction in our experiments. For **CoreSet** (Sener & Savarese, 2018), since the MIP optimized version is not scalable to large datasets, we use the time-efficient greedy approximation version by choosing

the node closest to the cluster centers. To adapt **AGE** (Cai et al., 2017) for the one-shot setting, we stick to the default parameters in the original work ($\gamma = 0.3$ for Citeseer, $\gamma = 0.7$ for Cora and $\gamma = 0.9$ for Pubmed. For benchmarks not mentioned, we set $\gamma = 0.8$. For methods that depend on K-Medoids, we apply an efficient approximation by (Park & Jun, 2009) by initializing with K-Means++ (Arthur & Vassilvitskii, 2007) and selecting nodes closest to centers.

## 5.2 Experiment Results on GCN

We provide the active learning results of GCN on each dataset in Figure 2. We further provide the exact average Macro-F1 scores and their standard errors in Table 2. For Ogbn-Arxiv, we reported both the Macro and Micro (Accuracy) F1 scores. Overall, we observe in Figure 2 and Table 2 that the proposed methods, GraphPart and GraphPartFar, outperform baseline methods in a wide range of budgets before the performance saturates. On smaller datasets where the number of classes is relatively small, the proposed methods outperform baseline methods by a large margin under a moderate budget size. On Corafull and Arxiv where the number of classes is much larger, the proposed methods demonstrate advantages over baseline methods (in particular, FeatProp) in a slightly later stage. This may be due to that some classes were not yet fully visited so the model has not been learning reliably when the budget size is small. Indeed, as shown in Figure 2d, 3a, the difference between our methods and FeatProp is not statistically significant until the budget size is at least 160. As can be seen in Tables 3 on Ogbn-Arxiv benchmark, no statistically significant difference can be observed from the Micro-F1 evaluation among the active learning methods when the budget is larger than 320, but our methods significantly outperformed baselines on Macro-F1.

Table 2: Summary of the performance of GCN on each benchmark. The numerical values represent the average Macro-F1 score of 10 independent trials and the error bar denotes the standard error of the mean (all in %). For Ogbn-Arxiv, we also include the Micro-F1 score (accuracy) as standard practice. The **bold** marker denotes the best performance and the underlined marker denotes the second-best performance. Asterisk (*) means the difference between our strategy and the best baseline strategy is statistically significant by a pairwise t-test at a significance level of 0.05.

| Baselines | Cora | | | Citeseer | | | Pubmed | | |
|---|---|---|---|---|---|---|---|---|---|
| Budget | 20 | 40 | 80 | 10 | 20 | 40 | 10 | 20 | 40 |
| Random | $49.7 \pm 9.7$ | $60.7 \pm 8.7$ | $71.6 \pm 5.0$ | $28.4 \pm 12.6$ | $37.6 \pm 6.7$ | $48.9 \pm 5.8$ | $49.1 \pm 11.4$ | $55.7 \pm 10.6$ | $69.5 \pm 6.2$ |
| Uncertainty | $37.6 \pm 8.2$ | $55.2 \pm 10.0$ | $70.1 \pm 6.4$ | $18.8 \pm 7.1$ | $24.2 \pm 5.7$ | $42.8 \pm 12.5$ | $46.0 \pm 11.5$ | $54.7 \pm 10.4$ | $64.8 \pm 9.2$ |
| Density | $49.8 \pm 8.6$ | $58.7 \pm 5.5$ | $73.1 \pm 4.1$ | $22.6 \pm 7.5$ | $37.8 \pm 9.0$ | $47.5 \pm 6.4$ | $43.6 \pm 9.8$ | $52.1 \pm 8.0$ | $68.7 \pm 5.3$ |
| CoreSet | $50.6 \pm 6.2$ | $64.1 \pm 5.3$ | $73.4 \pm 3.3$ | $27.2 \pm 6.6$ | $36.3 \pm 7.8$ | $49.6 \pm 7.5$ | $45.3 \pm 8.5$ | $55.7 \pm 13.0$ | $66.3 \pm 8.6$ |
| Degree | $57.7 \pm 0.6$ | $61.1 \pm 1.3$ | $74.4 \pm 0.5$ | $19.1 \pm 0.1$ | $28.0 \pm 0.3$ | $38.5 \pm 0.3$ | $53.9 \pm 1.9$ | $51.2 \pm 0.9$ | $54.4 \pm 0.7$ |
| Pagerank | $35.1 \pm 0.9$ | $60.2 \pm 1.2$ | $70.8 \pm 0.6$ | $11.7 \pm 0.3$ | $33.0 \pm 0.5$ | $45.6 \pm 0.9$ | $45.5 \pm 0.6$ | $51.0 \pm 0.6$ | $66.4 \pm 0.1$ |
| AGE | $50.3 \pm 5.0$ | $66.7 \pm 4.1$ | $74.2 \pm 2.7$ | $20.9 \pm 6.3$ | $35.5 \pm 9.2$ | $45.2 \pm 7.7$ | $47.2 \pm 9.9$ | $65.2 \pm 8.3$ | $70.3 \pm 8.0$ |
| FeatProp | $\underline{71.0} \pm 5.7$ | $76.1 \pm 2.5$ | $79.9 \pm 0.9$ | $27.9 \pm 5.5$ | $42.9 \pm 4.5$ | $53.7 \pm 4.5$ | $59.1 \pm 5.5$ | $65.4 \pm 5.2$ | $\underline{75.1} \pm 2.8$ |
| GraphPart | $\mathbf{76.1}^* \pm 2.7$ | $\underline{78.1} \pm 1.5$ | $\underline{80.3} \pm 1.6$ | $\mathbf{45.0}^* \pm 0.7$ | $\underline{45.4} \pm 4.1$ | $\mathbf{59.0}^* \pm 2.0$ | $\underline{63.0} \pm 0.7$ | $\mathbf{73.2}^* \pm 1.0$ | $74.9 \pm 1.3$ |
| GraphPartFar | $68.6 \pm 1.5$ | $\mathbf{78.1} \pm 2.1$ | $\mathbf{82.0}^* \pm 0.7$ | $\underline{35.1}^* \pm 0.6$ | $\mathbf{55.2}^* \pm 2.4$ | $\underline{57.5}^* \pm 2.9$ | $\mathbf{75.7}^* \pm 0.3$ | $\underline{67.5} \pm 0.5$ | $\mathbf{76.2} \pm 0.9$ |

| Baselines | Co-CS | | | Co-Physics | | | Corafull | | |
|---|---|---|---|---|---|---|---|---|---|
| Budget | 20 | 40 | 80 | 10 | 20 | 40 | 160 | 320 | 640 |
| Random | $43.8 \pm 8.8$ | $58 \pm 7.7$ | $76.1 \pm 6.3$ | $65.6 \pm 14.3$ | $73.1 \pm 7.5$ | $82.5 \pm 7.6$ | $23.8 \pm 2.0$ | $33.2 \pm 1.8$ | $43.2 \pm 2.1$ |
| Uncertainty | $32.4 \pm 8.0$ | $51.6 \pm 6.2$ | $66.7 \pm 6.4$ | $48.1 \pm 12.6$ | $56.3 \pm 9.3$ | $75.8 \pm 9.2$ | $17.3 \pm 1.7$ | $28.1 \pm 2.1$ | $39.1 \pm 1.2$ |
| Density | $34.2 \pm 8.0$ | $49.1 \pm 4.6$ | $60.1 \pm 6.0$ | $53.8 \pm 8.0$ | $67.9 \pm 12.0$ | $72.4 \pm 8.8$ | $21.3 \pm 1.1$ | $29.0 \pm 0.9$ | $38.3 \pm 1.1$ |
| CoreSet | $32.6 \pm 4.3$ | $50.3 \pm 4.8$ | $63.4 \pm 7.2$ | $43.8 \pm 15.3$ | $56.2 \pm 14.4$ | $79.8 \pm 8.3$ | $22.8 \pm 1.4$ | $33.4 \pm 1.1$ | $43.7 \pm 0.5$ |
| Degree | $31.1 \pm 0.5$ | $49.8 \pm 0.5$ | $53.8 \pm 0.1$ | $13.4 \pm 0.0$ | $13.4 \pm 0.0$ | $13.4 \pm 0.0$ | $20.3 \pm 0.7$ | $28.4 \pm 0.6$ | $37.6 \pm 0.6$ |
| Pagerank | $52.9 \pm 0.5$ | $67.6 \pm 1.5$ | $78.5 \pm 0.5$ | $82.8 \pm 0.4$ | $78.2 \pm 0.2$ | $84.9 \pm 0.2$ | $22.5 \pm 0.4$ | $29.9 \pm 0.7$ | $39.7 \pm 0.5$ |
| AGE | $39.3 \pm 6.6$ | $67.8 \pm 8.0$ | $77.9 \pm 5.0$ | $62.1 \pm 10.6$ | $75.6 \pm 7.9$ | $85.9 \pm 3.8$ | $22.2 \pm 1.4$ | $32.6 \pm 1.1$ | $43.0 \pm 0.9$ |
| FeatProp | $\underline{60.6} \pm 3.6$ | $77.2 \pm 3.2$ | $84.5 \pm 0.6$ | $79.7 \pm 1.9$ | $\mathbf{88.6} \pm 1.7$ | $\mathbf{90.6}^* \pm 0.4$ | $\underline{29.6} \pm 1.0$ | $37.6 \pm 0.8$ | $\underline{46.7} \pm 0.8$ |
| GraphPart | $\mathbf{68.5}^* \pm 0.2$ | $\mathbf{78.9} \pm 2.7$ | $\mathbf{86.2}^* \pm 1.2$ | $\mathbf{84.8}^* \pm 0.2$ | $86.1 \pm 0.1$ | $89.3 \pm 0.2$ | $\mathbf{31.0}^* \pm 1.3$ | $\mathbf{41.2}^* \pm 1.4$ | $\mathbf{48.6} \pm 0.5$ |
| GraphPartFar | $56.8 \pm 0.7$ | $\underline{77.4} \pm 1.1$ | $\underline{84.6} \pm 1.3$ | $\underline{81.2} \pm 0.3$ | $\underline{87.6} \pm 0.4$ | $\underline{90.1} \pm 0.3$ | $28.0 \pm 1.2$ | $\underline{38.4} \pm 0.6$ | $44.3 \pm 0.7$ |

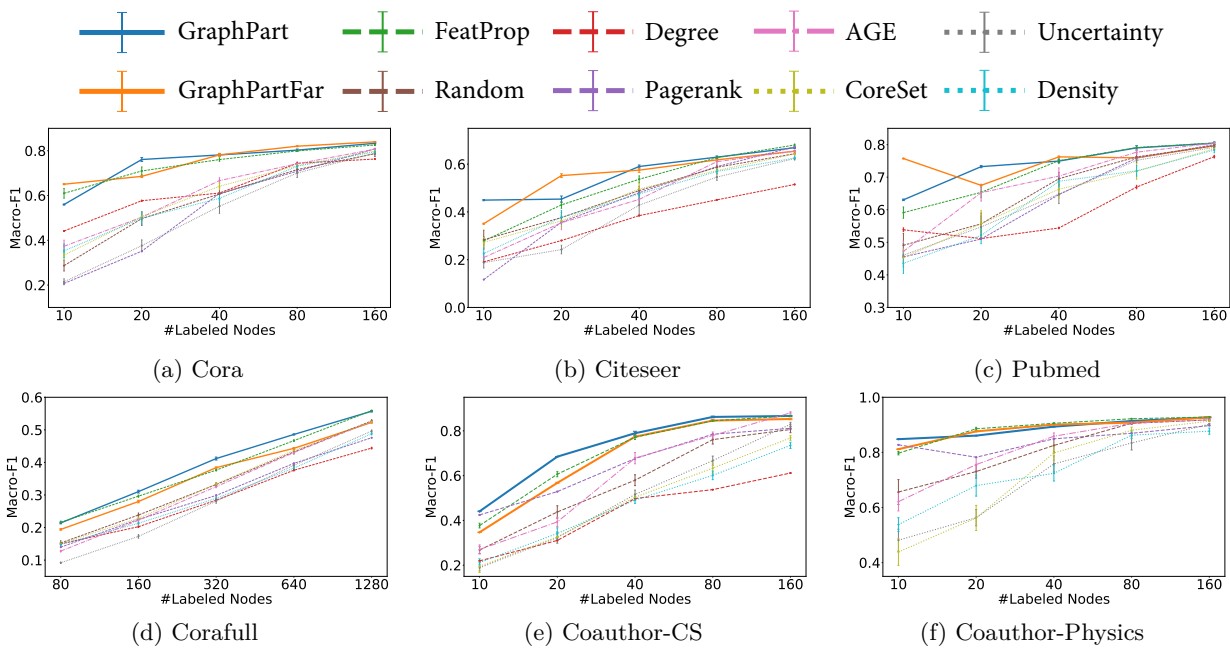

Figure 2: Performance of GCN by budget, with training set selected by each active learning baseline in one shot, averaged from 10 different runs. Our methods are **embolden**. In each sub-figure, the Macro-F1 score is plotted against the budget in a logarithm scale.

Table 3: Summary of the performance of GCN on Ogbn-Arxiv (Table 2 Continued). We also include the Micro-F1 score (accuracy) as standard practice.

| Baselines | Ogbn-Arxiv (Macro-F1) | | | | Ogbn-Arxiv (Micro-F1) | | | |
|---|---|---|---|---|---|---|---|---|
| Budget | 160 | 320 | 640 | 1280 | 160 | 320 | 640 | 1280 |
| Random | $21.9 \pm 1.4$ | $27.6 \pm 1.5$ | $33.0 \pm 1.4$ | $37.2 \pm 1.1$ | $52.3 \pm 0.8$ | $56.4 \pm 0.8$ | $60.0 \pm 0.7$ | $63.5 \pm 0.4$ |
| Uncertainty | $17.5 \pm 1.2$ | $24.3 \pm 1.8$ | $30.5 \pm 1.0$ | $36.7 \pm 0.9$ | $44.5 \pm 3.0$ | $51.7 \pm 1.0$ | $58.0 \pm 1.2$ | $62.5 \pm 0.4$ |
| Density | $16.3 \pm 1.6$ | $20.7 \pm 1.4$ | $24.8 \pm 0.7$ | $29.5 \pm 1.1$ | $46.3 \pm 2.3$ | $51.8 \pm 1.4$ | $56.0 \pm 1.1$ | $60.2 \pm 0.5$ |
| CoreSet | $19.5 \pm 1.8$ | $25.5 \pm 1.2$ | $30.5 \pm 1.2$ | $35.0 \pm 0.9$ | $47.6 \pm 2.9$ | $53.2 \pm 1.4$ | $58.1 \pm 0.8$ | $61.3 \pm 0.3$ |
| Degree | $8.0 \pm 0.2$ | $12.0 \pm 0.4$ | $12.0 \pm 0.3$ | $15.4 \pm 0.3$ | $29.2 \pm 0.9$ | $42.1 \pm 0.5$ | $42.9 \pm 0.9$ | $48.5 \pm 0.5$ |
| Pagerank | $21.8 \pm 0.6$ | $28.9 \pm 0.1$ | $34.0 \pm 0.2$ | $38.2 \pm 0.1$ | $50.8 \pm 1.6$ | $\underline{56.8} \pm 0.1$ | $\mathbf{60.8} \pm 0.2$ | $\mathbf{64.2} \pm 0.0$ |
| AGE | $20.4 \pm 0.9$ | $25.9 \pm 1.1$ | $31.7 \pm 0.8$ | $36.4 \pm 0.8$ | $48.3 \pm 2.3$ | $54.9 \pm 1.6$ | $60.0 \pm 0.7$ | $63.5 \pm 0.3$ |
| FeatProp | $24.0 \pm 0.7$ | $28.5 \pm 0.6$ | $33.6 \pm 0.4$ | $39.1 \pm 0.8$ | $51.9 \pm 1.1$ | $56.5 \pm 0.6$ | $60.4 \pm 0.4$ | $63.5 \pm 0.3$ |
| GraphPart | $24.2 \pm 0.7$ | $\underline{29.5}^* \pm 0.8$ | $\mathbf{36.4}^* \pm 0.5$ | $\mathbf{41.0}^* \pm 0.5$ | $\underline{52.3} \pm 0.9$ | $\mathbf{56.8} \pm 0.8$ | $\underline{60.7} \pm 0.6$ | $\underline{63.6} \pm 0.5$ |
| GraphPartFar | $\mathbf{26.0}^* \pm 0.7$ | $\mathbf{30.2}^* \pm 0.6$ | $\underline{36.1}^* \pm 0.7$ | $\underline{40.8}^* \pm 0.7$ | $\mathbf{53.8}^* \pm 1.0$ | $56.3 \pm 0.5$ | $59.4 \pm 0.3$ | $63.5 \pm 0.4$ |

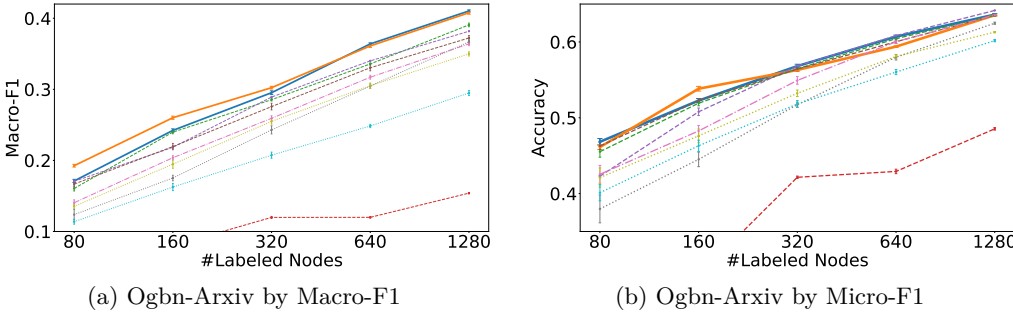

Figure 3: Performance of GCN on Ogbn-Arxiv (Figure 2 Continued).

## 5.3 Generalizability to Other GNNs

We further the experiment results on other GNN architectures (GraphSAGE and GAT) in Tables 4, 5 and Figure 4, 5 (Appendix). Overall, the superior performance of the proposed methods, especially against the state-of-the-art baseline FeatProp, demonstrates that selecting training nodes within proper partitions of the graph significantly helps the active learning performance.

Table 4: Summary of the performance of GraphSAGE. All the markers follow Table 2

| Baselines | Cora | | | Citeseer | | | Pubmed | | |
|---|---|---|---|---|---|---|---|---|---|
| Budget | 20 | 40 | 80 | 10 | 20 | 40 | 10 | 20 | 40 |
| Random | $36.0 \pm 9.6$ | $49.1 \pm 8.5$ | $64.2 \pm 5.6$ | $24.5 \pm 11.9$ | $32.4 \pm 6.6$ | $46.1 \pm 5.6$ | $40.6 \pm 13.0$ | $52.3 \pm 12.2$ | $66.3 \pm 7.9$ |
| Uncertainty | $23.5 \pm 5.0$ | $38.3 \pm 10.6$ | $58.6 \pm 7.8$ | $17.6 \pm 6.3$ | $25.1 \pm 6.1$ | $35.7 \pm 4.4$ | $35.2 \pm 6.6$ | $50.5 \pm 10.1$ | $64.1 \pm 10.5$ |
| Density | $38.0 \pm 8.3$ | $47.3 \pm 6.2$ | $62.1 \pm 5.4$ | $22.4 \pm 9.1$ | $35.1 \pm 4.9$ | $49.5 \pm 4.7$ | $32.7 \pm 8.9$ | $51.2 \pm 10.1$ | $62.1 \pm 9.9$ |
| CoreSet | $28.7 \pm 5.4$ | $47.8 \pm 9.4$ | $64.1 \pm 9.2$ | $16.6 \pm 5.2$ | $28.2 \pm 7.4$ | $39.0 \pm 7.4$ | $32.9 \pm 7.8$ | $53.1 \pm 10.2$ | $63.5 \pm 7.4$ |
| Degree | $43.0 \pm 1.5$ | $48.2 \pm 1.4$ | $65.4 \pm 2.1$ | $16.6 \pm 0.9$ | $23.1 \pm 1.1$ | $28.6 \pm 1.2$ | $39.7 \pm 0.5$ | $41.9 \pm 0.1$ | $43.0 \pm 0.1$ |
| Pagerank | $27.8 \pm 1.6$ | $47.9 \pm 1.2$ | $68.5 \pm 0.9$ | $13.0 \pm 1.0$ | $29.8 \pm 1.3$ | $38.3 \pm 2.2$ | $29.7 \pm 0.3$ | $41.7 \pm 0.6$ | $62.9 \pm 0.3$ |
| AGE | $38.2 \pm 8.0$ | $56.9 \pm 7.8$ | $69.5 \pm 2.7$ | $19.8 \pm 5.0$ | $29.7 \pm 4.9$ | $42.6 \pm 6.0$ | $39.9 \pm 15.4$ | $59.5 \pm 9.8$ | $70.2 \pm 4.7$ |
| FeatProp | $52.5 \pm 4.6$ | $61.4 \pm 5.5$ | $72.8 \pm 2.6$ | $23.4 \pm 4.3$ | $\underline{39.9} \pm 6.2$ | $\underline{53.3} \pm 3.8$ | $48.0 \pm 5.9$ | $59.1 \pm 6.0$ | $73.6 \pm 1.7$ |
| GraphPart | $\mathbf{61.4}^* \pm 4.7$ | $\mathbf{70.0}^* \pm 2.4$ | $\underline{76.2}^* \pm 2.7$ | $\mathbf{34.1}^* \pm 2.6$ | $36.1 \pm 6.4$ | $\mathbf{54.0} \pm 4.6$ | $\mathbf{52.0} \pm 0.8$ | $\mathbf{71.5}^* \pm 0.5$ | $\mathbf{74.6} \pm 1.1$ |
| GraphPartFar | $\underline{57.9}^* \pm 2.8$ | $\underline{69.7}^* \pm 4.2$ | $\mathbf{78.6}^* \pm 1.5$ | $\underline{30.7}^* \pm 2.3$ | $\mathbf{46.9} \pm 5.0$ | $53.1 \pm 4.0$ | $\underline{49.7} \pm 3.1$ | $\underline{70.7}^* \pm 1.6$ | $\underline{74.2} \pm 0.4$ |
| Baselines | Co-CS | | | Co-Physics | | | Corafull | | |
| Budget | 20 | 40 | 80 | 10 | 20 | 40 | 160 | 320 | 640 |
| Random | $38.3 \pm 8.7$ | $52.4 \pm 5.8$ | $69.7 \pm 5.2$ | $58.3 \pm 13.8$ | $66.9 \pm 10.1$ | $78.3 \pm 7.1$ | $15.4 \pm 1.1$ | $23.4 \pm 1.4$ | $33.5 \pm 1.3$ |
| Uncertainty | $36.5 \pm 6.3$ | $56.0 \pm 9.4$ | $72.5 \pm 3.5$ | $44.9 \pm 10.6$ | $50.6 \pm 8.4$ | $70.5 \pm 8.5$ | $15.3 \pm 0.8$ | $27.0 \pm 1.9$ | $39.7 \pm 1.1$ |
| Density | $31.5 \pm 8.5$ | $39.4 \pm 6.2$ | $54.3 \pm 3.3$ | $55.8 \pm 7.9$ | $56.8 \pm 11.5$ | $77.8 \pm 8.9$ | $13.0 \pm 1.2$ | $19.7 \pm 0.4$ | $30.6 \pm 1.0$ |
| CoreSet | $25.0 \pm 6.5$ | $40.8 \pm 4.3$ | $55.0 \pm 5.3$ | $39.0 \pm 14.1$ | $50.7 \pm 11.3$ | $75.6 \pm 9.5$ | $12.2 \pm 1.2$ | $19.7 \pm 0.6$ | $31.2 \pm 0.7$ |
| Degree | $23.1 \pm 5.2$ | $33.1 \pm 5.1$ | $44.9 \pm 3.0$ | $13.4 \pm 0.0$ | $13.4 \pm 0.0$ | $13.4 \pm 0.0$ | $10.6 \pm 0.4$ | $16.7 \pm 0.7$ | $26.8 \pm 0.7$ |
| Pagerank | $50.6 \pm 1.3$ | $59.4 \pm 1.8$ | $75.5 \pm 0.7$ | $\mathbf{81.0}^* \pm 0.8$ | $75.6 \pm 0.7$ | $82.6 \pm 0.4$ | $12.4 \pm 0.3$ | $19.4 \pm 0.8$ | $30.3 \pm 0.4$ |
| AGE | $36.5 \pm 6.3$ | $56.0 \pm 9.4$ | $72.5 \pm 3.5$ | $63.7 \pm 7.8$ | $71.0 \pm 8.8$ | $82.4 \pm 3.9$ | $13.5 \pm 0.8$ | $21.4 \pm 0.8$ | $33.1 \pm 1.0$ |
| FeatProp | $\underline{55.9} \pm 3.2$ | $68.6 \pm 3.2$ | $74.1 \pm 1.7$ | $77.0 \pm 2.4$ | $85.2 \pm 2.2$ | $\mathbf{90.0} \pm 0.7$ | $\underline{18.7} \pm 0.8$ | $25.8 \pm 0.7$ | $\underline{35.0} \pm 1.6$ |
| GraphPart | $\mathbf{62.9}^* \pm 1.4$ | $\underline{71.8}^* \pm 2.2$ | $\underline{80.3}^* \pm 1.3$ | $75.8 \pm 2.7$ | $\underline{85.8} \pm 0.3$ | $\underline{88.9} \pm 0.3$ | $\mathbf{19.7}^* \pm 0.9$ | $\mathbf{28.3} \pm 0.7$ | $\mathbf{36.1}^* \pm 1.0$ |
| GraphPartFar | $51.9 \pm 1.9$ | $\mathbf{73.3}^* \pm 2.2$ | $\mathbf{81.5}^* \pm 2.4$ | $\underline{78.5} \pm 1.1$ | $\mathbf{87.6}^* \pm 0.9$ | $88.8 \pm 0.5$ | $17.5 \pm 1.0$ | $\underline{26.2} \pm 1.4$ | $34.1 \pm 0.9$ |

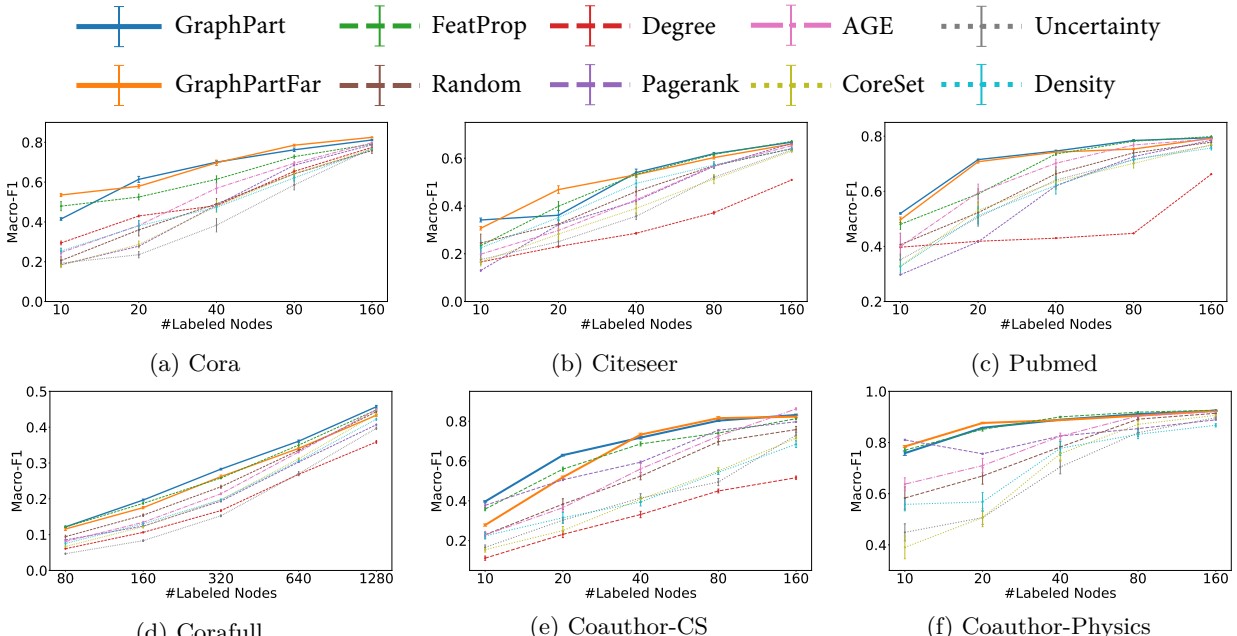

Figure 4: Results of each baseline on benchmark datasets on GraphSAGE averaged from 10 different runs.

Table 5: Summary of the performance of GAT. All the markers follow Table 2

| Baselines | Cora | | | Citeseer | | | Pubmed | | |
|---|---|---|---|---|---|---|---|---|---|
| Budget | 20 | 40 | 80 | 10 | 20 | 40 | 10 | 20 | 40 |
| Random | 41.2 ± 9.4 | 51.2 ± 8 | 62.3 ± 4.3 | 27.1 ± 11.4 | 35.9 ± 4.9 | 46.7 ± 5.3 | 41.6 ± 9.5 | 48.8 ± 11.6 | 64.5 ± 7.7 |
| Uncertainty | 23.2 ± 5.5 | 26.5 ± 4.9 | 47.2 ± 6.6 | 21.5 ± 6.7 | 25.8 ± 7.5 | 37.1 ± 8.0 | 43.9 ± 10.5 | 47.7 ± 11.2 | 56.7 ± 10.2 |
| Density | 41.6 ± 9.7 | 47.7 ± 5.6 | 62.8 ± 5.0 | 27.0 ± 9.0 | 36.7 ± 6.8 | 45.1 ± 6.8 | 48.4 ± 9.8 | 45.8 ± 13.5 | 60.1 ± 11.2 |
| CoreSet | 29.1 ± 6.2 | 44.9 ± 9.1 | 61.5 ± 6.9 | 19.6 ± 9.8 | 26.3 ± 7.9 | 44.3 ± 5.6 | 49.3 ± 12.9 | 54.6 ± 10.2 | 60.4 ± 13.0 |
| Degree | 39.3 ± 3.5 | 47.4 ± 2.3 | 61.3 ± 3.3 | 15.9 ± 1.2 | 20.0 ± 4.0 | 30.6 ± 2.0 | 44.3 ± 8.4 | 39.9 ± 4.3 | 40.2 ± 5.5 |
| Pagerank | 34.8 ± 4.1 | 50.3 ± 2.5 | 65.7 ± 1.2 | 15.4 ± 0.6 | 33.0 ± 2.4 | 42.5 ± 1.9 | 40.9 ± 13 | 46.3 ± 7.0 | 61.7 ± 3.7 |
| AGE | 39.8 ± 5.5 | 56.5 ± 6.3 | 66.7 ± 2.7 | 24.9 ± 12.5 | 32.9 ± 7.8 | 51.0 ± 5.0 | 42.0 ± 13.4 | 51.1 ± 9.7 | 64.8 ± 9.6 |
| FeatProp | 56.7 ± 5.8 | 58.7 ± 5.7 | 69.8 ± 2.1 | 24.9 ± 4.9 | 38.5 ± 7.9 | 50.4 ± 4.2 | 49.2 ± 8.1 | 58.4 ± 6.8 | 71.9 ± 2.4 |
| GraphPart | 59.1 ± 6.0 | 63.9* ± 3.5 | 71.8 ± 2.4 | 38.6* ± 4.3 | 38.1 ± 5.8 | 53.5 ± 4.0 | 63.0* ± 6.6 | 68.2 ± 2.2 | 68.4 ± 8.3 |
| GraphPartFar | 58.9 ± 2.9 | 66.7* ± 1.7 | 73.8* ± 2.2 | 33.6* ± 4.1 | 46.6* ± 3.3 | 52.2 ± 4.2 | 52.1 ± 12.4 | 60.6 ± 7.3 | 66.2 ± 2.8 |

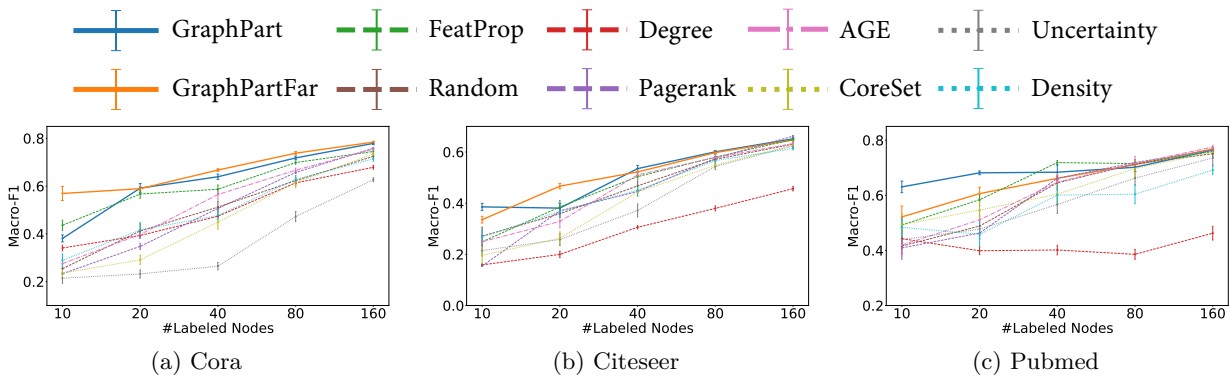

Figure 5: Results of each baseline on benchmark datasets on GAT averaged from 10 different runs.

## 5.4 Mitigating Accuracy Disparity of GNNs

The GNN models are reported to be less accurate on nodes that are further away from the training nodes, which leads to potential fairness concerns (Ma et al., 2021). As the goal of proposed active learning methods is to have the selected training nodes evenly distributed on the graph, we conduct an analysis on the accuracy disparity of the actively learned GNN models, following the study by (Ma et al., 2021).

We analyze the generalization performances by splitting the test nodes into 10 subgroups according to their aggregated-feature distance to the 40 selected training nodes (320 for CoraFull), and report the test accuracy on each subgroup. As can be seen in Figure 6, While the GNNs learned by GraphPart and GraphPartFar are still less accurate for test nodes that are further away, the proposed active learning methods are able to mitigate the accuracy disparity of GCN predictions compared to training with random training nodes in most datasets.

## 6 Conclusion and Discussion

In this work, we investigate the problem of active learning for GNNs. Inspired by the commonly seen local and global smoothness properties on graph-structured data, we propose a graph-partition-based active learning framework for GNNs, with two variants of concrete algorithms. The proposed framework can be seen as an active training node selection algorithm that approximately optimizes an upper bound of the expected classification error of unlabeled nodes. Through extensive experiments, we show that the proposed methods significantly outperform existing state-of-the-art baseline active learning methods. Furthermore, the proposed active learning methods are able to mitigate the accuracy disparity phenomenon commonly seen in GNN models.

Similar to previous work, *e.g.*, FeatProp, one limitation of this work is that their effectiveness relies on the smoothness assumptions to hold for the data of interest. Nevertheless, the proposed methods outperform

previous approaches on many benchmarks, which suggests that it is likely to generalize to more real-world applications.

**Broader Impact Statement**

As a potential positive impact, our proposed methods has demonstrated superior label efficiency. When applied to real-world applications, it can significantly reduce the economic cost of annotations in machine learning tasks on graphs.

For the potential negative impacts, the active learning procedure might introduce potential biases into the construction of training data set, which remains underexplored in the literature. In this work, we find that the proposed methods can actually reduce the accuracy disparity of GNN predictions compared to randomly selected training nodes. However, the consequences in terms of other fairness metrics, such as *equal opportunity*, still require dedicated further investigations. Therefore applications of active learning in high-stake scenarios should proceed with extra caution.

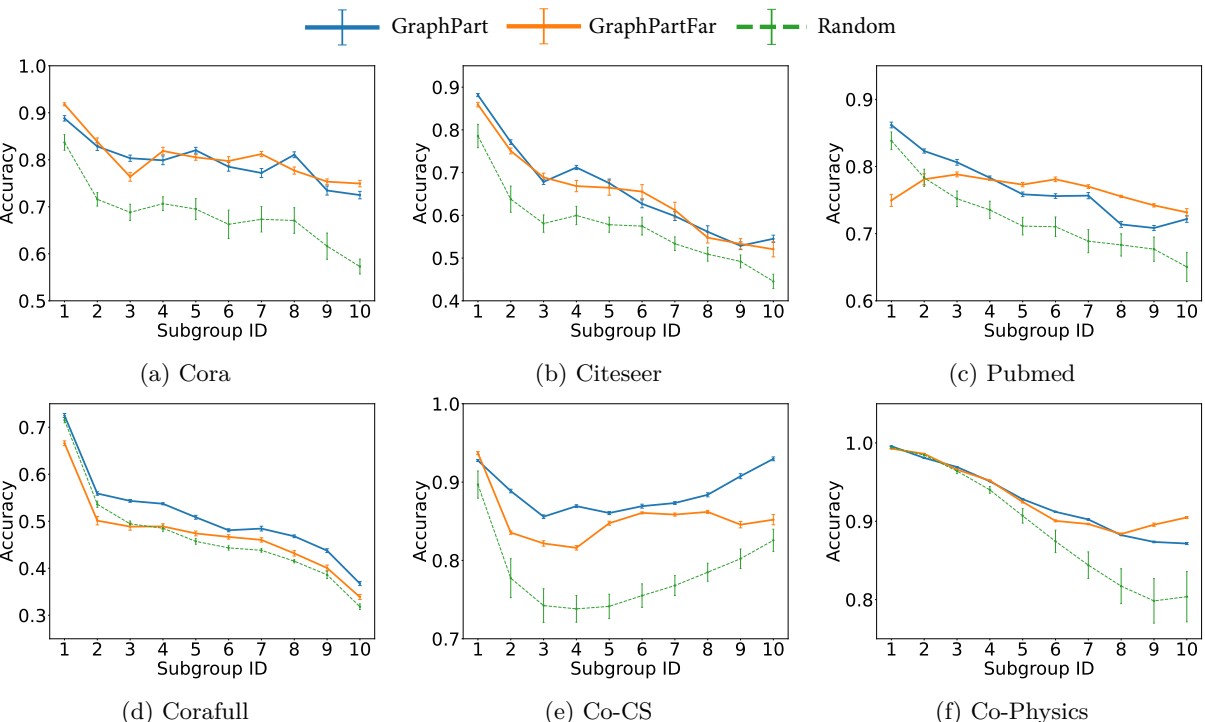

Figure 6: Accuracy disparity across 10 subgroups. Increasing subgroup indices represent the increasing distance to the selected training set.

**Acknowledgments**

This work was partially supported by the National Science Foundation under grant numbers 1633370 and 1949634. The authors would like to thank the anonymous reviewers for their helpful comments.

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

# A    Appendix

## A.1    Proof of Proposition 1

Before we start the proof of Proposition 1, we first introduce a helpful lemma below.

**Lemma 2.** *Under Assumption 2, for any $i \in S_{te}$, if $S_{tr} \cap T(i) \neq \emptyset$, letting $\tau(i) := \arg\min_{l \in S_{tr} \cap T(i)} \|g(v_i) - g(v_l)\|_2$ and $\varepsilon_i := \|g(v_i) - g(v_{\tau(i)})\|_2$, then we have*

$$||f(v_i) - f(v_{\tau(i)})||_\infty \leq \delta_h \varepsilon_i. \tag{5}$$

*Proof of Lemma 2.*

$$
\begin{aligned}
&||f(v_i) - f(v_{\tau(i)})||_\infty \\
&= ||h(g((v_i)) - h(g(v_{\tau(i)}))||_\infty \\
&\leq \delta_h ||g((v_i)) - g(v_{\tau(i)})||_2 &&(Assumption\ 2) \\
&= \delta_h \varepsilon_i.
\end{aligned}
$$

$\square$

We now start the proof of Proposition 1.

*Proof of Proposition 1.* We first consider the expected difference between the loss of $v_i$ and $v_{\tau(i)}$:

$$
\begin{aligned}
&\mathbb{E}_{y_i}[\mathcal{L}_0(f(v_i), y_i)] - \mathbb{E}_{y_{\tau(i)}}[\mathcal{L}_{\gamma_i}(f(v_{\tau(i)}), y_{\tau(i)})] \\
&= \sum_{c=1}^{C} P[y_i = c \mid v_i] \mathcal{L}_0(f(v_i), c) \\
&\quad - \sum_{c=1}^{C} P[y_{\tau(i)} = c \mid v_{\tau(i)}] \mathcal{L}_{\gamma_i}(f(v_{\tau(i)}), c) \\
&= \sum_{c=1}^{C} \eta_c(v_i) \mathcal{L}_0(f(v_i), c) - \sum_{c=1}^{C} \eta_c(v_{\tau(i)}) \mathcal{L}_{\gamma_i}(f(v_{\tau(i)}), c) \\
&= \textcolor{blue}{\sum_{c=1}^{C} \eta_c(v_i)\Big[\mathcal{L}_0(f(v_i), c) - \mathcal{L}_{\gamma_i}(f(v_{\tau(i)}), c)\Big]} \\
&\quad + \textcolor{red}{\sum_{c=1}^{C}\Big[\eta_c(v_i) - \eta_c(v_{\tau(i)})\Big]\mathcal{L}_{\gamma_i}(f(v_{\tau(i)}), c)}.
\end{aligned}
$$

By Assumption 1, the second term can be bounded as:

$$
\begin{aligned}
&\sum_{c=1}^{C}\Big[\eta_c(v_i) - \eta_c(v_{\tau(i)})\Big]\mathcal{L}_{\gamma_i}(f(v_{\tau(i)}), c) \\
&\leq \sum_{c=1}^{C}\Big|\eta_c(v_i) - \eta_c(v_{\tau(i)})\Big|\mathcal{L}_{\gamma_i}(f(v_{\tau(i)}), c) \leq \sum_{c=1}^{C}\Big|\eta_c(v_i) - \eta_c(v_{\tau(i)})\Big| \\
&\leq \sum_{c=1}^{C} \delta_\eta \Big|\Big|g(v_i) - g(v_{\tau(i)})\Big|\Big|_2 = \sum_{c=1}^{C} \delta_\eta \varepsilon_i = C\delta_\eta \varepsilon_i.
\end{aligned}
$$

We then prove that for $\gamma_i = 2\delta_h \varepsilon_i$, the first term is non-positive. We prove it by discussing two cases:

1. $\mathcal{L}_{\gamma_i}(f(v_{\tau(i)}), c) = 1$: In this case $\mathcal{L}_0(f(v_i), c) - \mathcal{L}_{\gamma_i}(f(v_{\tau(i)}), c) \leq 0$ always holds and the first term is non-positive.

2. $\mathcal{L}_{\gamma_i}(f(v_{\tau(i)}), c) = 0$: This situation only happens if $f(v_{\tau(i)})[c] > \gamma_i + \max_{b \neq c} f(v_{\tau(i)})[b]$. By Lemma 2, we know that for any $b = 1, \ldots, C$:

$$f(v_i)[c] \geq f(v_{\tau(i)})[c] - \delta_h \varepsilon_i$$

$$\text{and}\quad f(v_i)[b] \leq f(v_{\tau(i)})[b] + \delta_h \varepsilon_i$$

Therefore,

$$f(v_i)[c] - \max_{b \neq c} f(v_i)[b]$$
$$\geq \Big( f(v_{\tau(i)})[c] - \delta_h \varepsilon_i \Big) - \Big( \max_{b \neq c} f(v_{\tau(i)})[b] + \delta_h \varepsilon_i \Big)$$
$$= f(v_{\tau(i)})[c] - 2\delta_h \varepsilon_i - \max_{b \neq c} f(v_{\tau(i)})[b]$$
$$= f(v_{\tau(i)})[c] - \gamma_i - \max_{b \neq c} f(v_{\tau(i)})[b] > 0,$$

which implies $\mathcal{L}_0(f(v_i), c) = 0$. Hence $\mathcal{L}_0(f(v_i), c) - \mathcal{L}_{\gamma_i}(f(v_{\tau(i)}), c) = 0$ and the first term is non-positive.

Therefore, for $\gamma_i = 2\delta_h \varepsilon_i$, we have

$$\mathbb{E}_{y_i}[\mathcal{L}_0(f(v_i), y_i)] \leq C \delta_\eta \varepsilon_i + \mathbb{E}_{y_{\tau(i)}}[\mathcal{L}_{2\delta_h \varepsilon_i}(f(v_{\tau(i)}), y_{\tau(i)})].$$

$\square$

## A.2   Addendum to Experiments

We further conduct a sensitivity analysis for the proposed method. In particular, we are interested in the effectiveness of the partition approach when combined with distance metrics on different node representations. Specifically, (1) **Aggregation**: aggregated node features $S^2 X$; (2) **Embedding**: the last hidden layer of GCN trained on one-third of the budget; and (3) **Feature**: the original node features. As is shown in Table 6, the graph-partition step is robustly effective when combined with various types of distance metrics. For each choice of distance, we evaluate the active learning performance with or without the graph-partition step. We perform the experiments on Citeseer, Pubmed, and Cora. On each dataset, we evaluate each active learning method with a budget size of 20, 40, and 80. Each experiment is repeated with 10 random seeds.

As can be seen from Table 6, the graph-partition step is robustly effective when combined with various types of distance metrics. This suggests that the proposed graph-partition framework may be able to generalize to active learning for other graph representation learning methods. It is also interesting to observe that methods with a graph-partition step tend to have lower standard errors.

Table 6: Summary of the performance of different combinations of distance metric on Citeseer, Pubmed and Cora datasets with or without using graph partition. The numerical values represent the average Macro-F1 score of 10 independent trials and the error bar denotes the standard error of the mean (all in %). The **bold** marker denotes the better performance between with and without using graph partition, and asterisk (*) means this difference is statistically significant by a pairwise t-test at a significance level of 0.05.

| Node Rep. | With | Cora | | | Citeseer | | | Pubmed | | |
|---|---|---|---|---|---|---|---|---|---|---|
| Budget | Partition | 20 | 40 | 80 | 20 | 40 | 80 | 20 | 40 | 80 |
| Aggregation | yes | **76.1**\* ± 2.7 | **78.1** ± 1.5 | **80.3** ± 1.6 | **45.4** ± 4.1 | **59.0**\* ± 2.0 | **67.7** ± 1.5 | **73.2**\* ± 1.0 | 74.9 ± 1.3 | **79.7** ± 2.3 |
| | no | 71.0 ± 5.7 | 76.1 ± 2.5 | 79.9 ± 0.9 | 42.9 ± 4.5 | 53.7 ± 4.5 | 67.4 ± 2.0 | 65.4 ± 5.2 | **75.1** ± 2.8 | 79.5 ± 0.9 |
| Embedding | yes | **61.3**\* ± 4.8 | **69.4**\* ± 3.9 | **76.7** ± 3.4 | **43.9** ± 7.1 | **54.4** ± 2.8 | **61.1** ± 2.3 | **70.1** ± 8.8 | **74.9** ± 2.8 | **78.1** ± 2.0 |
| | no | 54.5 ± 4.7 | 62.6 ± 5.7 | 74.1 ± 3.7 | 38.2 ± 9.0 | 50.6 ± 5.1 | 59.7 ± 2.3 | 63.4 ± 10.6 | 70.3 ± 6.6 | 77.0 ± 1.5 |
| Feature | yes | **65.6**\* ± 2.6 | **71.0**\* ± 2.1 | 77.2 ± 1.3 | **15.7**\* ± 0.9 | **33.1** ± 3.4 | **54.4** ± 1.8 | 56.6 ± 2.1 | 67.1 ± 2.1 | **77.1** ± 2.0 |
| | no | 53.2 ± 5.2 | 64.0 ± 5.1 | **77.4** ± 1.7 | 6.1 ± 2.4 | 30.9 ± 4.5 | 54.0 ± 4.1 | **64.3**\* ± 8.6 | **70.0** ± 4.1 | 76.6 ± 0.9 |

