# OpenReview forum: "Partition-Based Active Learning for Graph Neural Networks"
_TMLR — Accepted by TMLR_

### Review · Reviewer_PWue · 2023-01-15

**Summary Of Contributions:**

In this paper, the authors proposed a partition-based active learning method called GraphPart for improving graph neural network predictions. The loss function is inspired by Proposition 1, which is obtained according to some existing results such as Sener& Savarese,2018 and Wu et al., 2019b.  Extensive experimental results on 7 datasets demonstrate the effectiveness of the proposed approach when compared with 8 baselines.

**Audience:**

Yes

**Broader Impact Concerns:**

Not applicable.

**Claims And Evidence:**

No

**Requested Changes:**

See my comments on weaknesses.

**Strengths And Weaknesses:**

Strengths:

1. This submission is clear and easy to follow.

2. The loss is motivated by theoretical analysis, which is reasonable.

Weaknesses:

1. In general, this paper does not give me an impressive feeling. Specifically, as the authors mentioned in Section 2, active learning on GNN has been explored in many existing works. The authors also mentioned using local and global smoothness to design the final learning objective, where global smoothness is largely missing in the literature on active learning for GNNs. However, it is not clear why this global smoothness is important. This motivation and significance should be further discussed in the method part and experimental part.

2. I am very confused about Proposition 1. First, $\epsilon$ is defined but not used in Eq. (2). In Appendix Lemma 2, $j(i)$ should be $\tau (i)$. Second, the authors mentioned 'the second term (the expected margin loss of $\tau (i)$) is increasing with respect to $\xi_i$.' However, it is hard for me to get this finding from Eq. (2). I suggest the authors provide more evidence to support this conclusion. In addition, how to get a conclusion about 'This upper bound motivates an active learning algorithm that selects the training set by minimizing $\sum_{i\in S_{te}} \xi_i$'? This should be further explained.

3. The proof of Lemma 1 is missing.

4. Some details are missing in experiments. For example, what are 20, 40, and 80 in Table 2 (for the Cora dataset)?

5. My another major concern is the loss function.  In your submission, the theoretical results are based on the 0-1 loss assumption in Eq. (1). However, I checked your code, and you use cross-entropy loss to do all experiments. Therefore, there is a large gap between theory and experiments. I expect the authors can explain the reasons.

---

> ### Author Response · Authors · 2023-01-28
> **Response to Reviewer PWue**
>
> We thank the reviewer for the fast review and great questions. We have addressed them by both responding in this post and revising our manuscript. However, **per the recommendation by TMLR, we are asked to defer the update of the manuscript after all three reviewers have posted the reviews**. So we have not updated our manuscript on OpenReview at the time of posting this response. Below is our detailed response to each question.
>
> ### Q1
>
> This is a great question. We add more explanations on the motivation for global smoothness. First, global smoothness is a commonly seen phenomenon in graph-structured data and is explicitly utilized in many conventional graph learning methods prior to the GNN era (Zhou et al., 2004; Gu & Han, 2012). Second, local smoothness usually only considers pairwise euclidean distances between the node representations. However, the euclidean distance becomes less informative in high-dimensional space [1]. Global smoothness is a great supplement to local smoothness.
>
> We have also added these explanations to our revised manuscript.
>
> [1] Aggarwal, Charu C., Alexander Hinneburg, and Daniel A. Keim. "On the surprising behavior of distance metrics in high dimensional space." In Database Theory—ICDT 2001: 8th International Conference London, UK, January 4–6, 2001 Proceedings 8, pp. 420-434. Springer Berlin Heidelberg, 2001.
>
> ### Q2
>
> Thanks for the clarification question. We address the three sub-questions in detail below.
>
> 1. The $\epsilon$ in proposition 1 is a typo: it should be $\varepsilon$ instead. And $j(i)$ in Lemma 2 should indeed be better changed to $\tau(i)$ for consistency. Thanks for spotting them!
> 2. The second term in Eq. (2) is a margin loss with margin $\gamma_i$ (see definition in Eq. (1)), which is increasing with respect to the margin $\gamma_i$ and hence is also increasing with respect to $\varepsilon_i$ (as $\gamma_i = 2\delta_h \varepsilon_i$).
> 3. As the whole upper bound is increasing with respect to $\varepsilon_i$'s, it motivates the objective of minimizing the sum of $\varepsilon_i$'s.
>
> We have also made these clarifications in our revised manuscript.
>
> ### Q3
>
> We omitted the proof of Lemma 1 as it is a quite straightforward result. The proof can be done in a couple of lines. Assuming Assumption 3 holds, since the inequality in Assumption 3 holds for every pair of (i, j), it will also hold for any pair of (i, j) restricted to a partition. Therefore, when $\delta_{\eta} = \delta_{\eta}'$, the conclusion of Assumption 1 holds. Therefore, there exists at least one $\delta_{\eta} \le \delta_{\eta}'$ (e.g., $\delta_{\eta} = \delta_{\eta}'$) such that Assumption 1 holds.
>
> Note that, however, the inverse of Lemma 1 does not hold, i.e., Assumption 1 does not imply Assumption 3. So Assumption 3 is a strictly stronger assumption than Assumption 1.
>
> We have now added this in the appendix for completeness.
>
> ### Q4
>
> The numbers 20, 40, 80 are the budget sizes. We will make it clear in the revised version and also proofread other similar parts.
>
> ### Q5
>
> We first note that 0-1 loss is difficult to optimize. So in practice, surrogate losses such as cross-entropy loss are used. Nevertheless, in statistical learning theory, it is still common to have the analysis on 0-1 loss while having empirical evaluations using surrogate losses. While we acknowledge that there is a gap between the theory and the experiments, we believe that the theory is a good proxy of reality given the common practice in learning theory.

---

> > ### Comment · Reviewer_PWue · 2023-02-05
> > **Response**
> >
> > Thank you for your response! My concerns are solved.

---

> > > ### Author Response · Authors · 2023-02-05
> > > **Thank you!**
> > >
> > > Thank you for your helpful feedback!

---

### Review · Reviewer_58wd · 2023-01-29

**Summary Of Contributions:**

This paper studies active learning for graph neural networks. However, the fundamental theories are not correct. As a result, all methods that are based on this theoretical result are not reasonable. Thus, this paper needs a major revision to fix this fatal error.

**Audience:**

Yes

**Broader Impact Concerns:**

Since there are some theoretical errors, it is no sense to discuss its broader impact.

**Claims And Evidence:**

No

**Requested Changes:**

There are some errors when proving Proposition 1.

**Strengths And Weaknesses:**

When bounding the second term in the proof of Proposition 1, the first step does not hold.

Considering the case $\eta_c\left(v_i\right)-\eta_c\left(v_{\tau(i)}\right)<0$ for $\forall c$, you cannot get the first step. Therefore, Proposition 1 is not true. Then, the proposed method is not reasonable.

---

> ### Author Response · Authors · 2023-01-29
> **Response to Reviewer 58wd**
>
> Dear Reviewer,
>
> That step indeed missed the absolute value sign. With a quick fix, the whole proof would be valid. Thanks for spotting the typo. Please verify the following updated derivations for bounding the second term:
>
> $\sum_{c=1}^C [ \eta_c(v_i) - \eta_c(v_{\tau(i)}) ] L_{\gamma_i}(f(v_{\tau(i)}), c) $
>
> $\le \sum_{c=1}^C \Big| \eta_c(v_i) - \eta_c(v_{\tau(i)}) \Big| L_{\gamma_i}(f(v_{\tau(i)}), c)$   *(note that $L_{\gamma_i}(f(v_{\tau(i)}), c)$ is nonnegative)*
>
> $ \le \sum_{c=1}^C \Big| \eta_c(v_i) - \eta_c(v_{\tau(i)}) \Big| $
>
> $\le \sum_{c=1}^C \delta_\eta \Big|\Big| g(v_i) - g(v_{\tau(i)}) \Big|\Big|_2 $
>
> $    =\sum_{c=1}^C \delta_\eta \varepsilon_i $
>
> $    = C \delta_\eta \varepsilon_i.$

---

### Review · Reviewer_NTPU · 2023-02-01

**Summary Of Contributions:**

This paper proposes GraphPart a novel approach to a novel partition-based active learning
approach for GNNs which is predicated on partitioning an input graph into disjoint sets and performs queries construction on the representation space of nodes on the aggregation function $g$.  The paper also has a theoretical result on the upper bound on the classification error under various smoothness assumptions. Finally, extensive experiments are conducted

**Audience:**

Yes

**Claims And Evidence:**

Yes

**Requested Changes:**

The biggest change or addition I would like to see is a detailed investigation of different strategies to create the partitions. In particular, trying some popular off-the-shelf graph coarsening algorithms would be a big plus.

Secondly, an investigation into the label smoothness assumptions would strengthen the paper.

**Strengths And Weaknesses:**

Strengths:

- Overall the motivation for this problem is clear and active learning with structured input data such as graphs is indeed a worthwhile problem to tackle. I also found the writing and presentation to be of mostly high quality---outside of a few minor details that I will outline below. The experiments also seem exhaustive and it's important to see the proposed approach working well. Finally, I also checked the proof quickly but it seems correct.


Weaknesses:
I have a few concerns about this paper that I list below.
- While the overall problem of active learning on graph data is interesting and well-motivated, the idea of partitioning the graph into disjoint sets is less developed. Indeed, there is a vast literature on how to do this but perhaps a more fundamental question one could ask is why do we want to do graph partitioning in the first place? For example, one could instead choose to do graph coarsening and consider each super node as its own disjoint partition. This would open up a huge variety of new methods that could be employed and this angle should be investigated a bit more. Specifically, how robust are the current results to the partitioning/graph coarsening strategy?

- The paper makes a series of assumptions about label and model smoothness. These seem mostly important to prove Proposition 1. For label-smoothness it feels like you can actually empirically check this for different graphs by constructing $L$-hop neighborhoods and see to what extent real-world graphs or more specifically the ones in the datasets considered for the experiments.

- I find the experimental settings to be a bit ad-hoc. The number of labeled nodes should be a percent of the training dataset size rather than an absolute number, otherwise, trends across datasets are misleading.

- I found the comment about Assumption 3 in relation to Assumption 1 a bit confusing. It seems obvious that in Assumption 1 $\delta$ can be smaller as it's a stronger assumption because we are operating on the partitions rather than the whole graph.

---

> ### Author Response · Authors · 2023-02-08
> **Response to Reviewer NTPU (Part 1)**
>
> We appreciate the reviewer for the thoughtful comments. Please see our detailed responses below.
>
> ### Q1
>
> This is a great question and it partly coincides with the first question by Reviewer PWue. The reason we want to do graph partitioning is that we want to utilize the *global smoothness* appearing in many graph datasets for better active learning.
>
> In our response to Reviewer PWue, we have further clarified the motivation for global smoothness and we copy it here for completeness of this response: "*First, global smoothness is a commonly seen phenomenon in graph-structured data and is explicitly utilized in many conventional graph learning methods prior to the GNN era (Zhou et al., 2004; Gu & Han, 2012). Second, local smoothness usually only considers pairwise euclidean distances between the node representations. However, the euclidean distance becomes less informative in high-dimensional space. The global smoothness is a great supplement to the local smoothness.*"
>
> Regarding graph partitioning vs graph coarsening, the purpose of graph coarsening is mainly to reduce the graph size while maintaining certain properties of the graph. Graph coarsening seems to be more commonly used in graph-level prediction tasks, instead of node-level prediction tasks as we focus on in this work. So we think graph partitioning suits our approach better.
>
> Regarding the choice of the graph partitioning strategy, we first acknowledge that not all graph partitioning strategies work equally well. For example, the vanilla Clauset-Newman-Moore method can lead to many very small partitions with a handful of nodes, which hurts the active learning approach. We fix this issue by merging small partitions through agglomerative hierarchical clustering, as described in our paper. However, we would like to emphasize that a key merit of the proposed approach is that the graph-partitioning procedure (Clauset-Newman-Moore with agglomerative hierarchical clustering) is completely **hyperparameter-free**, and the number of partitions for each dataset is automatically decided within the procedure. The proposed active learning approach performs well on all the benchmark datasets when **applying the exactly same partitioning procedure on each dataset**. This suggests that the partitioning procedure used in our approach is very likely to be able to robustly generalize to other real-world graph datasets. So overall, we believe this procedure already well serves the primary purpose of this work: developing an effective active learning approach.
>
> ### Q2
>
> Thank you for the question. We first clarify that exactly measuring the label-smoothness assumption requires access to the ground-truth label distribution, which is not possible. However, we agree that this assumption is relevant to some proxy metrics, such as Homophily Ratio as defined in [1]. A few prior studies have reported that many graph datasets indeed have a high homophily ratio, suggesting that the label-smoothness assumption holds on these datasets. For completeness, we've further reported the homophily ratio of all the datasets in Table 1 in the revised paper.
>
> [1] Zhu et al. Beyond homophily in graph neural networks: Current limitations and effective designs. (https://arxiv.org/abs/2006.11468)
> [2] Ma et al. Is Homophily a Necessity for Graph Neural Networks? (https://arxiv.org/abs/2106.06134)
>
> ### Q3
>
> We agree that the number of labeled nodes may not be comparable across datasets. However, we think that the percentage of nodes is nothing better than the number of nodes, as larger datasets do not always lead to "harder" tasks. For example, in the official data split by the Planetoid version [1] of Pubmed and Cora datasets, the training set sizes of Pubmed and Cora are respectively 60 and 140, despite the total size of Pubmed being about 7 times of Cora. We also note that prior work (Wu et al. 2019b) also used absolute numbers as budget sizes.
>
> In fact, to avoid misleading the readers for comparing trends across datasets, we intentionally list the budget sizes *repeatedly within each dataset* in all the result tables and have *separate result figures for different datasets*.
>
> [1] Yang et al. Revisiting Semi-Supervised Learning with Graph Embeddings. (https://proceedings.mlr.press/v48/yanga16.html)

---

> > ### Author Response · Authors · 2023-02-08
> > **Response to Reviewer NTPU (Part 2)**
> >
> > ### Q4
> >
> > Thank you for the question and please allow us to better clarify it. When a condition A implies another condition B, then the condition A is stronger than B as A is sufficient for B. Therefore, according to our Lemma 1, Assumption 3 is stronger than Assumption 1. More intuitively, Assumption 3 requires the smoothness condition to *hold on the entire graph*, while Assumption 1 only requires the smoothness condition to *hold within some partitions of the graph*. Our Assumption 1 allows the label distributions to have a sharp switch across different partitions/communities (which is more realistic), while Assumption 3 required by previous work does not allow such sharp switches. So the difference between the two assumptions is also significant in practice.
> >
> > We also added more clarifications on this in our revised draft.

---

### Decision · Action_Editors · 2023-03-11

**Recommendation:** Accept as is

**Comment:**

This paper proposes a graph partition method for semi-supervised learning on graph data. The proposed method is well motivated and supported by both theoretical and thorough empirical analysis. There was a typo in the proof, but it was fixed after the reviewer pointed it out.

The remaining concern from one reviewer is that the high-level idea of this paper is not very impressive, which I agree. However, given that this paper proposes a new method with good performance, as shown in the extensive experiments.

Thus, I would suggest acceptance of this paper. Since This paper reviews and compares with a bunch of exisiting methods, I would recommend a "survey certification".

**Audience:**

Yes

**Claims And Evidence:**

Yes